# Wearable Sensor Technology to Predict Core Body Temperature: A Systematic Review

**DOI:** 10.3390/s22197639

**Published:** 2022-10-09

**Authors:** Conor M. Dolson, Ethan R. Harlow, Dermot M. Phelan, Tim J. Gabbett, Benjamin Gaal, Christopher McMellen, Benjamin J. Geletka, Jacob G. Calcei, James E. Voos, Dhruv R. Seshadri

**Affiliations:** 1School of Medicine, Case Western Reserve University, Cleveland, OH 44106, USA; 2Sports Medicine Institute, University Hospitals Cleveland Medical Center, Cleveland, OH 44106, USA; 3Department of Orthopaedic Surgery, University Hospitals Cleveland Medical Center, Cleveland, OH 44106, USA; 4Sanger Heart and Vascular Institute, Atrium Health, Charlotte, NC 28204, USA; 5Gabbett Performance Solutions, Brisbane, QLD 4000, Australia; 6Centre for Health Research, University of Southern Queensland, Ipswich, QLD 4305, Australia; 7Institute of Health and Wellbeing, Federation University, Ballarat, VIC 3350, Australia; 8University Hospitals Rehabilitation Services and Sports Medicine, Cleveland, OH 44106, USA; 9Department of Biomedical Engineering, School of Engineering, Case Western Reserve University, Cleveland, OH 44106, USA

**Keywords:** wearable technology, core body temperature, heat stroke, exertional heat illness, physiological modeling, machine learning, sports medicine, occupational physiology, athlete management systems

## Abstract

Heat-related illnesses, which range from heat exhaustion to heatstroke, affect thousands of individuals worldwide every year and are characterized by extreme hyperthermia with the core body temperature (CBT) usually > 40 °C, decline in physical and athletic performance, CNS dysfunction, and, eventually, multiorgan failure. The measurement of CBT has been shown to predict heat-related illness and its severity, but the current measurement methods are not practical for use in high acuity and high motion settings due to their invasive and obstructive nature or excessive costs. Noninvasive predictions of CBT using wearable technology and predictive algorithms offer the potential for continuous CBT monitoring and early intervention to prevent HRI in athletic, military, and intense work environments. Thus far, there has been a lack of peer-reviewed literature assessing the efficacy of wearable devices and predictive analytics to predict CBT to mitigate heat-related illness. This systematic review identified 20 studies representing a total of 25 distinct algorithms to predict the core body temperature using wearable technology. While a high accuracy in prediction was noted, with 17 out of 18 algorithms meeting the clinical validity standards. few algorithms incorporated individual and environmental data into their core body temperature prediction algorithms, despite the known impact of individual health and situational and environmental factors on CBT. Robust machine learning methods offer the ability to develop more accurate, reliable, and personalized CBT prediction algorithms using wearable devices by including additional data on user characteristics, workout intensity, and the surrounding environment. The integration and interoperability of CBT prediction algorithms with existing heat-related illness prevention and treatment tools, including heat indices such as the WBGT, athlete management systems, and electronic medical records, will further prevent HRI and increase the availability and speed of data access during critical heat events, improving the clinical decision-making process for athletic trainers and physicians, sports scientists, employers, and military officers.

## 1. Introduction

Exertional heat-related illness (HRI) is increasing in incidence as global temperatures rise, with at least 9000 student athletes and 2000 military members affected in the US each year [1,2,3,4]. The spectrum of illness of HRI ranges from heat-induced muscle cramps to life-threatening heat stroke, and a rising core body temperature (CBT) during exercise directly leads to HRI and a subsequent decline in performance [5,6]. Early detection and intervention are critical to improving HRI outcomes, as mortality can drop from 80% in those with delayed cooling to 0% when elevated CBT is detected and reduced to less than 40 °C within 30 min of symptom onset [7,8,9,10,11,12]. Football players, soccer players, long-distance runners, military servicemembers, outdoor and manual workers, and emergency responders are all frequently affected by HRI-related morbidity and mortality (Figure 1) [2,13,14,15,16,17,18,19]. Despite the frequency and potentially fatal consequences of undetected HRI, coaches, trainers, officers, and employers currently rely on visible cues and subjective assessments of their athletes and employees for early detection, such as malaise, confusion, thirst, ataxia, or excessive sweating. This can tragically delay recognition and treatment [7,9,20].

The monitoring of CBT can be instrumental in managing internal and external workloads to prevent the elevation of CBT to the critical temperatures at which HRI occurs and performance declines. A healthy resting CBT is considered to be 37 °C ± 0.5 °C or 98.7 ± 0.9 °F [21], and a slightly elevated CBT of around 38.5 °C/101.3 °F is a normal physiological response to exercise and generally not cause for concern [5,6,22]. Once CBT reaches 40 °C/104 °F, clinical diagnosis and treatment of HRI begins whether or not symptoms are present (Figure 2) [20]. CBT can rise as quickly as 1 °C every 5 min at near-peak exercise intensities due to the heat generated from skeletal muscle contraction, increased metabolic rate, and increased heart rate (HR) [5,22]. Hot and humid climates worsen the rise in CBT and risk for HRI [11], especially if the patient is not acclimatized to exercising in a harsh environment [6].

Wearable technology is currently playing a role in actively monitoring the physiological status and preventing injury in athletes at all levels, and so, the integration of low-power, high-fidelity predictive analytics with wearable sensors can enable the monitoring of CBT to ultimately provide the early detection and prevention of HRI [23,24,25,26,27]. Current CBT measurement methods during exercise are limited to esophageal, rectal, or telemetric gastrointestinal pill thermometers, which are various combinations of invasive, obstructive to actively exercising athletes, slow to respond to changing CBT, and excessively costly for routine use [28,29]. Other CBT measurement methods used in clinical practice, such as oral, tympanic, axillary skin, and temporal skin, have shown to be unreliable in exercising subjects [30]. Due to these challenges, there has been a significant effort to develop algorithms that provide accurate estimations of CBT by integrating key physiologic parameters such as heart rate, skin temperature, and skin heat flux (Table 1) [29,30,31,32,33,34].

CBT has thus far proven challenging to predict via algorithms and noninvasive wearable sensors due to the influence of a myriad of personal, situational, and environmental variables on individual rates of heat accumulation and dissipation. Factors such as gender, age, medication usage, BMI, level of heat acclimatization, exercise workload, sport and position, clothing level, ambient temperature, and ambient humidity all have shown a correlation to an individual’s CBT [22,36]. The risk of HRI is increased in hot and humid environments due to inefficiencies in evaporative cooling mechanisms, but CBT can also reach critical levels in more temperate environments secondary to prolonged high-intensity activity or wearing heavy clothing and equipment that independently prevent effective evaporative cooling [6,11,17,22,28,37,38,39,40,41]. Due to this variability in both individual heat tolerance and the environments in which people experience heat stress, CBT prediction algorithms must be validated on a wide variety of subjects and conditions. 

Machine learning (ML) platforms applied to CBT prediction algorithms may provide the opportunity to increase the accuracy of CBT predictions, thereby potentially decreasing the risk of HRI. ML methods have quickly become widespread in medicine and sports science, with applications for reduction of the injury burden already in use in baseball, basketball, soccer, American football, and Australian football [51]. ML can locate and utilize patterns in the data that are difficult or impossible for a human to identify, especially in large, multimodal data sets, and higher predictive power can be expected with increased data input [52]. Causal modeling or dimensionality reduction can be used to distinguish related variables [53,54], which could simplify the process of algorithm development by easily identifying which sensors and at what locations are most correlated with CBT.

The body of literature investigating CBT prediction systems is now growing, and algorithms are becoming more complex. Few of these scientifically evaluated algorithms using wearable devices are being used in practice to reduce the burden of HRI in at-risk populations. Yet, the number of wearable devices that report CBT measurements are increasing on the market. This systematic review provides a comprehensive understanding of how the prediction of CBT using wearable technology has been studied and the accuracy, reliability, and conditions in which CBT predictions have been validated in the literature (e.g., subject characteristics, exercise conditions, study methodology, and generalizability to larger populations). The results disseminating from this review will highlight how CBT is currently being calculated to better understand the gap in translational compatibility of this research into practical systems to prevent HRI and optimize performances in high-risk populations.

## 2. Materials and Methods

A systematic literature search was conducted in the Web of Science Core Collection database to identify works from 1 January 2000 until 31 December 2021 in which predicted CBT was compared to measured CBT in exercising subjects. This database was selected for its inclusion in both medical and engineering journals. The search used the terms below, where TS stands for topic and searches the title, abstract, author keywords, and Web of Science Keywords Plus for the specified term or terms.
TS = (core body temperature OR core temperature OR deep body temperature) AND TS = (noninvasive OR non-invasive OR wearable OR indirect) AND TS = (exercise* OR hot* OR heat* OR physical*) AND TS = (model* OR predict* or estimate*)

Studies were excluded if they met any of the following criteria: non-medical or non-human research;systematic and narrative reviews or meta-analyses that did not develop a previously unreported CBT prediction algorithm;did not include a wearable device (defined below);did not include exercising subjects;validated the CBT prediction model with the same data on which it was developed;did not use a sufficient comparator method (ingestible telemetric temperature pills, rectal thermometer, or esophageal thermometer);did not predict or measure CBT.

The study selection process and enumeration of CBT prediction algorithms are depicted in Figure 3. 

For this review, wearables were defined as epidermal patches, wrist monitors, and chest straps [54]. Sufficient comparators were defined as ingestible telemetric temperature pills, rectal thermometers, or esophageal thermometers in accordance with the previous literature demonstrating the effectiveness of these methods in measuring the core body temperature [30].

The following data were extracted from papers meeting the inclusion and exclusion criteria:Study metrics and methodology, including participant demographics and testing conditions;Characteristics of the CBT predictive algorithm, including modeling method and number and locations of sensors;Outcome measures of algorithm prediction accuracy, defined as the root mean square error (RMSE), mean difference (MD), or standard error of the estimate (SEE).

Studies were also classified as whether they collected prospective data or retrospective data. The testing conditions were recorded based on the variability of the exercise environment and the exercises performed. Studies in which the testing environment was considered variable included a difference of at least 2.0 °C and/or 10% relative humidity within a testing period or between separate testing periods. Studies in which the exercise intensity or pattern was considered variable between exercises must include at least 2 separate exercises in which the exercise performed, exercise intensity, or duration or sequence of the exercise periods differed.

Each individual algorithm reported and validated by a study was recorded as a separate data point in this review. In cases where values such as age and the RMSE were reported based on differing testing conditions but utilizing the same algorithm, a grand mean of that value was calculated and reported with weighting based on the number of subjects in each condition. Only data from wearables and participants used in the validated algorithm were included. Data from participants or trials used to train or develop algorithms were excluded. 

The primary outcome measure collected from the selected studies was the root mean square error or deviation (RMSE or RMSD), a commonly used metric for measuring the goodness-of-fit of a prediction algorithm. The RMSE can be interpreted as the standard deviation of the unexplained variance between the predicted and actual values. It is an absolute measure of fit calculated using the square root of the variance of the residuals and is thus expressed in the same units (°C) as the compared values. A lower RMSE value indicates a better fit [55]. When reported, ± 1 standard deviation (SD) of the RMSE was also recorded. 

Where the RMSE was not reported by a study, the mean difference (MD) or standard error of the estimate (SEE) were collected as the outcome measures, along with their respective measures of variability.

## 3. Results

### 3.1. Article Search Results

The systematic literature search identified 303 potential studies for analysis, 20 of which reported novel algorithms for CBT predictions using wearable sensors in exercising subjects that met all the inclusion criteria and none of the exclusion criteria. From those 20 study records, a total of 25 CBT prediction algorithms were identified, because 3 of the studies reported multiple algorithms. The root mean square error (RMSE) was reported as an outcome measure in 18 of the 25 algorithms, compared to the mean difference (MD) reported for 5 out of the 25 algorithms and standard error of the estimate (SEE) for 3 out of the 25 algorithms. The data used for validation of the algorithms were collected prospectively with the validation of the algorithm for 20 algorithms, while 6 of the algorithms were validated with retrospective data and one algorithm using both (Table 2).

### 3.2. Aggregate Results

A total of 592 unique subjects were used to validate the analyzed algorithms, with 211 subjects being used to validate more than one algorithm. The average sample size used to validate an algorithm was 32.1 ± 43.2 subjects, and the average age and mass of the subjects were 26.4 ± 4.7 years and 76.6 ± 4.4 kg, respectively (Table 2). The largest sample size used to validate an algorithm was 166 subjects [56], and the smallest sample size was 4 subjects [29]. All studies that reported the sex of their participants included males, with 7 out of 25 algorithms also reporting female subjects in validation. Variable exercise conditions (temperature and relative humidity differences greater than 2.0 °C and/or 10%) were reported in 20 out of 25 algorithms. Variable exercise sequences or intensities were reported in 16 out of 25 algorithms. The telemetric gastrointestinal thermometer pill was the most used comparator device, reported in 17 out of 25 algorithms, followed by a rectal thermometer was reported in 12. Several algorithms were validated using multiple different comparator methods. 

Due to the much lower number of algorithms reporting the MD or SEE as compared to the RMSE, and the inability to directly compare these measures to the RMSE, subsequent analyses were performed solely on algorithms reporting the RMSE. In addition, the outlier data from Tsadok et al. DW was excluded from further analysis due to the extremely high RMSE compared to all other data points.

### 3.3. Results from Studies Reporting RMSE

The unweighted average RMSE in the 18 algorithms in which it was reported was 0.38 °C, and the average standard deviation of the RMSE among those algorithms was 0.23 °C. The unweighted average RMSE decreased to 0.28 °C if a single outlier (1.97 °C in Tsadok et al. DW) was removed, and the average RMSE SD similarly decreased to 0.14 °C if the same outlier (1.26 °C in Tsadok et al. DW) was removed. A total of four algorithms reporting the RMSE utilized HR sensors alone to predict CBT, with the average RMSE and SD reported as 0.32 °C ± 0.15 °C across 139 subjects (Table 3). Algorithms using additional sensor modalities, such as skin heat flux, combined with HR reported an average RMSE and SD of 0.25 °C ± 0.13 °C in nine algorithms and 472 subjects. There were three algorithms reviewed that did not include HR, with an average RMSE and SD of 0.29 °C ± 0.15 °C across 39 subjects (Table 3). The most common wearable device location was the chest, with eight algorithms including only a wearable at the chest and two algorithms including a wearable at the chest in addition to devices at other locations. Algorithms with only a wearable at the chest had an average RMSE and SD of 0.29 °C ± 0.14 °C across 322 subjects, while those with a wearable at the chest plus at least one other device reported an average 0.32 °C ± 0.13 °C with 184 subjects. No wearable at the chest was included in five of the algorithms, and the average RMSE of these algorithms was 0.26 °C ± 0.15 °C across 81 subjects. A single wearable device was reported in 11 algorithms, with an average value of 0.27 °C ± 0.15 °C across 269 subjects. Multiple device locations were used in four algorithms with an average RMSE reported as 0.29 °C ± 0.12 °C with 354 subjects. The greatest degree of accuracy (lowest RMSE value) was observed in an algorithm that used a single wearable at the wrist, reporting a 0.13 °C RMSE in 15 subjects [46]. The greatest degree of accuracy (lowest RMSE value) was observed in an algorithm that used a single wearable at the wrist, reporting a 0.13 °C RMSE in 15 subjects [46] (Figure 4). One study did not report the location of the sensors used in its algorithm (Table 4).

### 3.4. Results from Studies Reporting MD or SEE

The mean difference (MD) was reported for 5 out of the 25 algorithms while the standard error of the estimate (SEE) was reported for 3 out of the 25 algorithms. The lowest MD, i.e., greatest degree of accuracy, was seen in the proprietary ML model using one wearable at the chest at 0.16. The lowest SEE, i.e., greatest degree of accuracy, was seen in an algorithm using bootstrap modeling with a neck wearable at 0.2 (Table 5).

## 4. Discussion

This systematic review investigated the wearable devices and algorithms used in CBT prediction today to better understand the accuracy and reliability of the existing methodologies. In addition, this review identified areas where future research and developments should be directed to improve the translatability of this technology to practical HRI prevention strategies. Our results show that multiple wearable devices ranging from wristwatches to ear sensors are being used in CBT prediction algorithms that utilize steady-state Kalman filters, regression, and ML methods such as Leave-One-Out Cross-Validation (LOOCV). A high accuracy was observed among published algorithms, with 17 out of 18 algorithms meeting a previously reported clinical acceptance threshold of a RMSE less than or equal to 0.5 °C [32,33,37,60]. The average RMSE of the 18 algorithms reporting this value in this report was 0.28 ± 0.14 °C after removal of the outlier, Tsadok et al. DW, which reported an RMSE of 1.97 °C or nearly five times the next-highest reported RMSE, likely due to its reliance on a single, distal measurement site, single sensor, and proprietary algorithm. This outlier was the only algorithm reporting an RMSE above 0.5 °C. These critical findings demonstrate that accurate CBT predictions using wearable devices is achievable under controlled, prescribed conditions, but algorithms must be validated in a wide range of conditions and subjects. Increasing the number of wearable devices worn did not appear to correlate with the increased accuracy in this review (RMSE 0.29 ± 0.15 °C in 233 subjects for single-device algorithms versus 0.29 ± 0.12 °C in 354 subjects for multiple-device algorithms; Table 4), showing that sufficient data can be collected for CBT prediction from a single wearable device. 

High accuracy across variable conditions and subjects was observed in the two algorithms in this review that reported the use of ML methods [31,57]. The first algorithm included ambient humidity and temperature sensors with skin temperature and heart rate in a single wearable device, the KENZEN, along with user-input individual characteristics, including age, mass in kilograms, and biological sex [31]. This algorithm showed a high degree of accuracy across variable conditions and exertion levels ranging from 13.4 to 43.2 °C and 32 to 110% predicted max HR (RMSE = 0.30 °C) when tested on 27 subjects [57]. The study used LOOCV to evaluate and compare the accuracy of multiple ML algorithms. The other algorithm using ML methods in this review used a proprietary ML method with the CORE device to predict CBT from the HR, skin temperature, and heat flux in consistent environmental conditions. This study reported the lowest MD and thus highest accuracy of the algorithms reporting that measure but could not be compared against the algorithms reporting the RMSE. In addition, this study did not validate the algorithm in varying environmental conditions or include environmental variables in the algorithm [57]. It also must be noted that, as a proprietary device and algorithm, the exact methods being used are unknown; thus, transparency in methods, development, accuracy, and reliability is critical to using such devices for clinical decision-making. While these studies showed immense promise for the use of ML methods to predict CBT due to the high accuracy observed, accurate predictions in controlled clinical and laboratory settings and in similar subjects may not translate to the variable conditions of a dynamic external environment [30,36]. Thus, the clinical validation of these algorithms in variable environments and larger, more diverse subject pools is still required.

The above studies using ML methods reflected similar limitations to others in this review, namely the lack of validity in dynamic real-world environments and lack of diverse subject pools. Even though 21 out of 25 total algorithms in this review conducted validations in at least two different environmental conditions (ambient humidity and temperature), many of these validations were in laboratory settings rather than in the field. Additionally, all but eight algorithms limited their validation testing to one clothing level [31,34,37,63], and the clothing level was included as a variable in only one algorithm [37]. Environmental conditions were included as variables in only three CBT prediction algorithms reporting the RMSE [31,37,46], including the above-mentioned algorithm using the KENZEN device, and these three algorithms reported a much more accurate average RMSE than algorithms not including environmental variables (0.18 °C across 105 subjects compared to 0.42 °C across 558 subjects). Additionally, the rates of heat accumulation and dissipation and, in turn, individual CBT have shown to be affected by individual variables, including gender, age, level of physical fitness, skin surface to body mass ratio, hydration status, sleep deprivation, and phase of the menstrual cycle [36]. Studies have also shown that the chances of HRI are increased by up to 40% in those with previous episodes of HRI; skin conditions increasing heat storage; cardiac or thyroid conditions preventing proper thermoregulation; those experiencing alcohol or opioid withdrawal; and those taking medications that affect the body’s ability to thermoregulate, including anticholinergics, antipsychotics, antihistamines, and stimulants [9,10,28,35]. However, the subject pools in the reported studies did not show much variability, with the average subject age and mass ranging from 21.6 to 31.1 years and 72.1 to 81.0 kg, only a third of the reviewed algorithms including female subjects in addition to male and most studies standardizing the results by excluding participants with medical conditions or taking medication. Only one algorithm included subject characteristics or demographics as variables [31]. While the reported results were accurate within their constrained conditions and populations, future algorithms must be validated across varied environments and populations at high risk for HRI, including athletes, soldiers, first responders, and those working physically intensive occupations. Ideally, algorithms will also be validated across both normal and pathologic CBTs, although subject safety dictates stopping exercise once reaching 40 °C due to the high risk of harm at this temperature. There remains a need to validate algorithms for high school athletes under 19 years of age, as this demographic makes up the largest proportion (nearly 50%) of HRI presenting to emergency rooms [68]. 

ML methods offer vast opportunities to address some of the current issues with CBT prediction with the wearable devices described in this review, namely the individualization of predictions, wearable device count, and interoperability with the existing HRI prevention tools. This systematic review demonstrated the ability of the current algorithms to predict CBT using wearable devices accurately under controlled conditions, as well as in varied conditions when including environmental variables such as in the algorithms in Moyen et al., Welles et al. and Nazarian et al. [29,31,46]. While the accuracy from the study is not in question, future studies should strive to validate CBT prediction algorithms using wearable devices in dynamic, real-world environments, such as in Moyen et al. and in a wider variety of subjects at risk for HRI. Rather than limiting the scope of validation conditions, future algorithms using ML methods may be able to personalize their results by including additional data and variables on individual subjects into future algorithms, thereby better controlling for the inherent variability between users. Additionally, ML methods such as causal modeling and dimensionality reduction, used to determine the correlations between variables, can help reduce the number of devices worn and eliminate redundant data sources, as demonstrated in the study by Eggenberger et al., which showed the ability to acquire highly similar correlations between two CBT prediction algorithms using either 18 unique measurement locations or by reducing their algorithms to include only 2 of those measurements (r^2^ = 0.70 versus r^2^ = 0.68, respectively) [60].

Most importantly, the interoperability and integration between wearable CBT prediction algorithms and preexisting tools for injury prevention will decrease the athletic trainer burden and provide a wholistic platform for expedited clinical decision-making during times of distress (Figure 5). Existing systems such as EMRs, heat indices such as the wet bulb globe temperature (WBGT), and the integration of CBT prediction algorithms into a holistic platform will provide team physicians, athletic trainers, head coaches, sports scientists, employers, and military officers an integrative platform to ultimately reduce the incidences of HRI and injury burden more broadly while optimizing performances. The algorithm could also integrate with the existing tools used in HRI prevention for athletes and employees, such as the WBGT, which estimates environmental heat stress by accounting for environmental variables, including the air temperature, humidity, wind speed, and thermal radiation [69], to estimate the effects of external influences on a subject’s heat status. This integrated system could also help prevent the under- or overuse of injuries and maximize performances by identifying problematic patterns and informing coaches, trainers, and officers on decisions to stop or alter exercises and keep athletes within an optimal temperature range to prevent HRI and maximize performances [47]. The future of interoperability between wearable CBT prediction algorithms using ML and the preexisting tools for injury prevention such as EMRs, heat indices, AMSs and worker management systems, and treatment algorithms will increase the availability and speed of access of important data for decision-making, thus preventing injuries and improving care at the clinic, athletic field, and battlefield. 

## 5. Conclusions

This systematic review focused on the current state of wearable sensor technology and predictive analytics towards monitoring the core body temperature in athletes, with applications in other fields of work in high acuity environments or situations (Figure 6). The findings of this review demonstrate the opportunity that exists for the development of accurate and versatile wearable sensors that integrate CBT monitoring algorithms to accurately predict the onset of and prevent heat-related illness. The overall volume of currently published literature on the application of wearable technology to monitor the core body temperature is limited, with only 20 studies and 25 CBT prediction algorithms for exercising subjects identified in this systematic review. Of the algorithms reporting a RMSE, 17 out of 18 algorithms reported here met the previously identified clinical validity standards, with an RMSE lower than 0.5 °C. The validation of these algorithms in variable populations and in real-world situations is still needed. A large opportunity exists for ML methods to incorporate individual and environmental characteristics, such as ambient temperature and subject demographics, into CBT prediction algorithms. Improved CBT prediction methods using ML will more reliably predict and prevent HRI across diverse environments and populations, but the future of CBT prediction lies in its integration with the larger HRI and athletic injury prevention space, which includes heat indices such as the WBGT, athlete management systems, electronic medical records, treatment algorithms, and RTP protocols. ML algorithms, with their increased data processing abilities, are ideal candidates for interoperability within these larger systems, as they can both supply input to the CBT prediction algorithm and use its output, along with other data, to assist in clinical decision-making, increasing the amount of data available to clinicians and streamlining transitions of care.

## Figures and Tables

**Figure 1 sensors-22-07639-f001:**
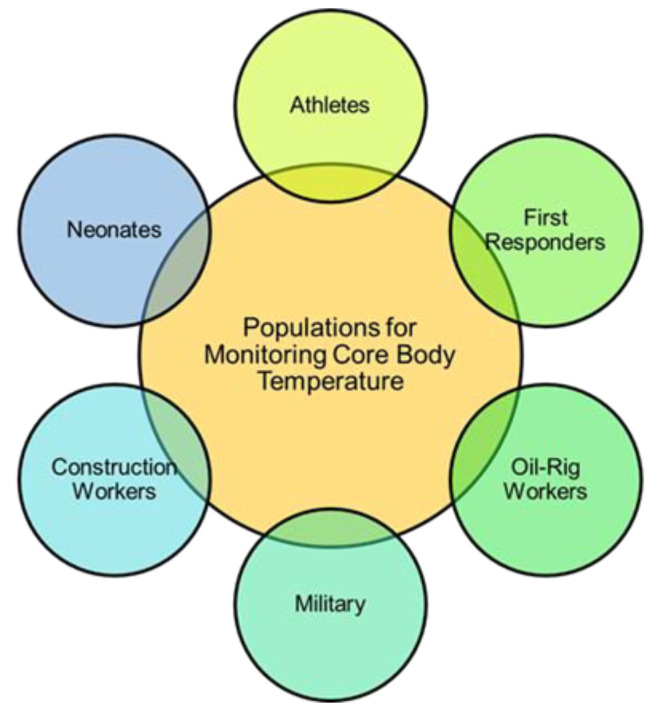
Populations at a high risk for HRI and, thus, most important for the monitoring of CBT. This review focuses on the monitoring of CBT in athletes and other patients in dynamic environments such as first responders, oil rig and construction workers, and military, as solutions already exist for inpatient and neonate monitoring.

**Figure 2 sensors-22-07639-f002:**
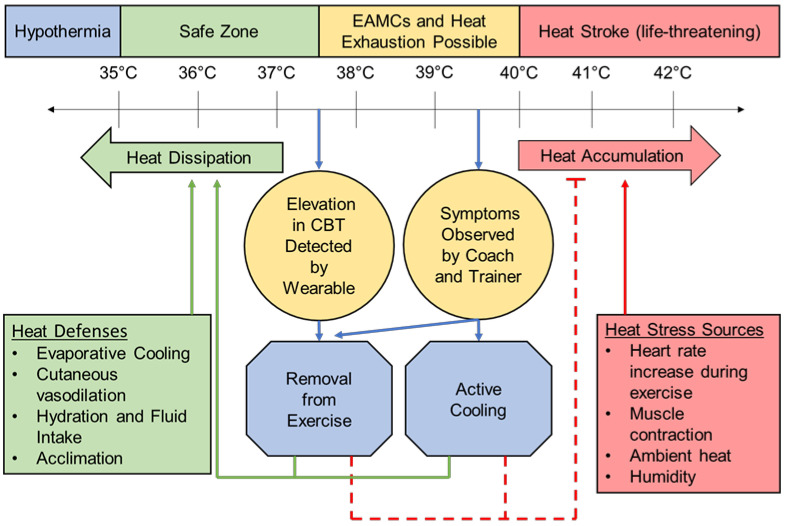
A schematic representation of the CBT at which heat-related illness occurs [5,6,20,21,22,35]. Includes how wearable CBT measurement systems can provide early intervention to reduce the need for the treatment and, ultimately, morbidity and mortality of HRI. EAMCs = exercise-associated muscle cramps.

**Figure 3 sensors-22-07639-f003:**
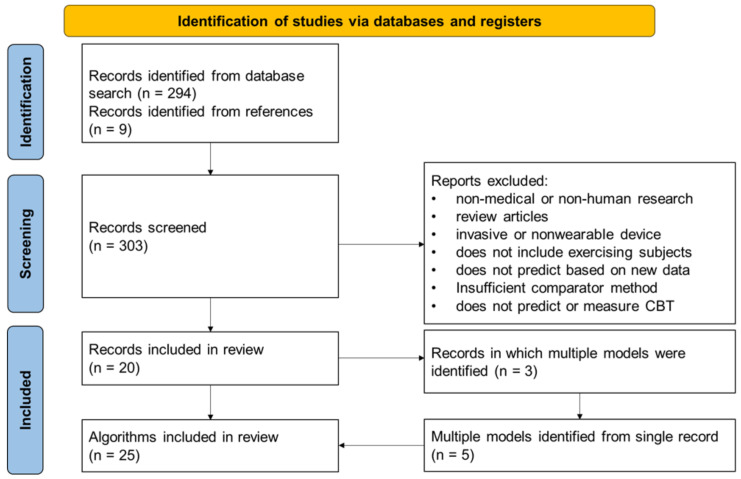
Flowchart depicting the search results and methodology of the study selection. Additionally depicted is the enumeration of multiple prediction models from some studies.

**Figure 4 sensors-22-07639-f004:**
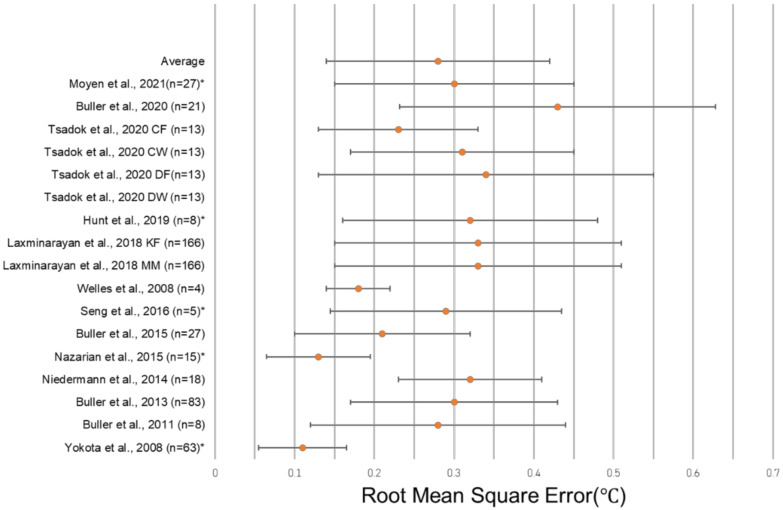
Forest plot depicting the RMSE ± standard deviation of studies reporting this metric. Tsadok et al. DW [47] is excluded both from the plot and from the displayed average due to its status as an outlier. The asterisks (*) denote models that did not report a RMSE SD, and so, the error bars represent ± ½ of the reported RMSE. [29,31,32,34,37,46,56,58,59,62,64,66].

**Figure 5 sensors-22-07639-f005:**
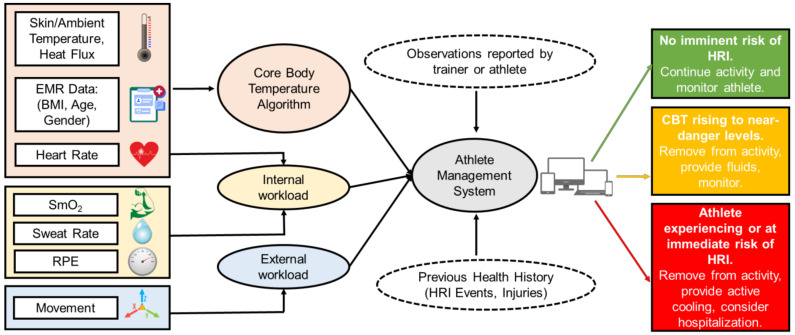
A workflow schematic depicting the hypothetical interoperability of an algorithm for CBT prediction using wearable devices [25,26,27,54]. Both sensor inputs to the CBT prediction algorithm and its output to other systems are included in the schematic. Heart Rate is used to calculate both the CBT and internal workload. EMR = Electronic Medical Record, RPE = Rating of Perceived Exertion, and SmO_2_ = Muscle Oxygen Saturation.

**Figure 6 sensors-22-07639-f006:**
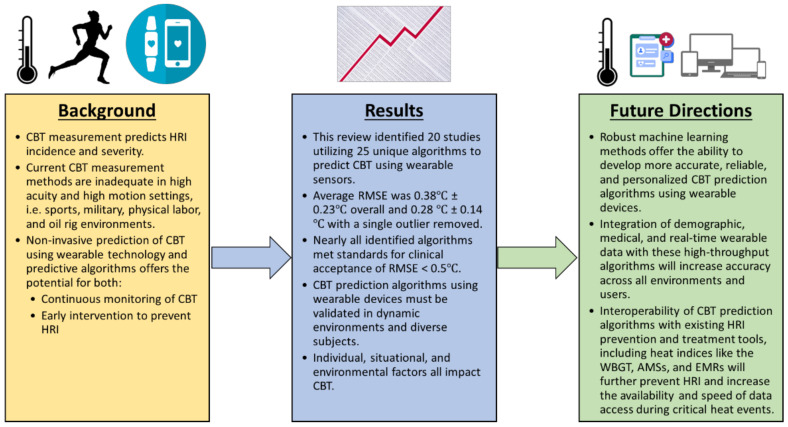
Conclusions from this review and proposed future directions for the technology. HRI = heat-related illness, CBT = core body temperature, RMSE = root mean square error, WBGT = wet bulb globe temperature, AMS = athlete management system, and EMR = electronic medical record.

**Table 1 sensors-22-07639-t001:** Comparison of methods used to measure CBT in exercising subjects. Summarized from reference [30]. Commercial devices listed are only those marketed or validated as measuring exercising CBT.

Method	Commercial Devices in Use for Exercising CBT Measurement	Advantages	Disadvantages
Gastrointestinal	HQ, Inc. CorTemp [42]	Accurate measurement of internal CBT Results are easily obtained even in athletic environment	Requires repeated ingestion High cost and requires receiver device Injured athletes who need an MRI and have ingested device cannot be imaged
Rectal	Henry Schein DataTherm II [43] YSI Temperature Sensor [43] WelchAllyn SureTemp Plus [43]	Accurate measurement of internal CBT Standard of Care Cost-efficient	Uncomfortable/painful Not robust enough for athletic use
Esophageal	Medtronic Mon-A-Therm [44] Circa S-Cath Temperature Probe [45]	Accurate measurement of internal CBT	Highly sensitive to small changes in CBT—readings constantly changing Uncomfortable/unreasonable in athletic settings
Oral	None	Results are easily obtained Standard of Care and readily available Cost-efficient	Not an accurate measurement of CBT Not capable of continuous monitoring
Temporal	None	Quick temperature readings Capable of continuous monitoring	Inaccurate readings Requires sensors placed on face Could inhibit athletic protective equipment (e.g., helmets)
Wearable Sensors (e.g., wrist monitors)	Nazarian et al. [46] Tsadok et al. CW [47] Tsadok et al. DW [47]	Built for athletic use without impeding athlete movement Capable of continuous monitoring	Not as accurate as rectal monitoring Efficacy has yet to be validated during athletic activities
Epidermal Sensors	Moyen et al. [31] greenTEG CORE [48] Equivital EQ02 [49] Zephyr BioHarness [50]	Potential for extremely comfortable athletic use Integration into textiles worn by athletes	Efficacy has yet to be validated during athletic activities Not as accurate as rectal monitoring

**Table 2 sensors-22-07639-t002:** Methodological and demographic details of the included studies. The clothing level was recorded as follows: light for short sleeve athletic wear, heavy for long sleeves and pants, and occupational for military or other specialized clothing. PC = Prospectively Collected and RC = Retrospectively Collected data. NR = Not Reported, V = Variable, O = Occupational Gear, H = Heavy (long sleeves and pants), and L = light (short sleeves and shorts).

Model	Year	Validation Data Source	Number of Validation Subjects, *n*	Average Subject Age	Average Subject Mass (kg)	Subject Sex	Subject Clothing Level	Variable Exercise Conditions	Variable Exercises	CBT Comparator
Moyen et al. [31]	2021	PC	27	28.9	75.2	M, F	V	Y	Y	GI Pill, Rectal
Nazarian et al. [46]	2021	PC	15	NR	NR	M, F	NR	N	N	GI Pill
Verdel et al. [57]	2021	PC	25	30	77.9	M	NR	Y	Y	Rectal
Buller et al. [58]	2020	PC	21	21	80.5	M	O	N	N	GI Pill
Tsadok et al. CF [47]	2020	PC	13	25	72	M	O	Y	N	Rectal
Tsadok et al. CW [47]	2020	PC	13	25	72	M	O	Y	N	Rectal
Tsadok et al. DF [47]	2020	PC	13	25	72	M	O	Y	N	Rectal
Tsadok et al. DW [47]	2020	PC	13	25	72	M	O	Y	N	Rectal
Hunt et al. [59]	2019	PC	8	26.4	77.4	M	O	Y	Y	GI Pill
Eggenberger et al. MAX [60]	2018	PC	6	28.9	74.1	M	V	N	Y	Rectal
Eggenberger et al. MIN [60]	2018	PC	6	28.9	74.1	M	V	N	Y	Rectal
Laxminarayan et al. KF [56]	2018	RC	166	23.2	76.5	M, F	V	Y	Y	Rectal, Esophageal
Laxminarayan et al. MM [56]	2018	RC	166	23.2	76.5	M, F	V	Y	Y	Rectal, Esophageal
Welles et al. [29]	2018	PC	4	22	76.4	NR	O	Y	Y	GI Pill
Mazgaoker et al. [61]	2017	PC	17	24.16	67.8	M	L	Y	N	Rectal
Seng et al. [62]	2016	RC	5	NR	NR	M	NR	Y	Y	GI Pill
Seo et al. [63]	2016	PC	27	22.3	78.1	M	V	Y	Y	Rectal
Buller et al. [64]	2015	PC	27	29.9	82.5	M	O	Y	Y	GI Pill
Niedermann et al. [32]	2014	RC, PC	18	23.5	74.2	M	H	Y	Y	Rectal, GI Pill
Buller et al. [34]	2013	RC	83	22.9	81.6	M, F	V	Y	Y	GI Pill, Rectal
Richmond et al. [65]	2013	PC	32	38	81.9	NR	O	Y	Y	Rectal
Buller et al. [66]	2011	PC	8	27.7	85.7	M	O	N	Y	GI Pill
Teunissen et al. [67]	2011	PC	7	25.4	73.2	M, F	L	Y	N	Rectal
Gunga et al. [33]	2008	PC	20	39.5	83.5	M	O	Y	N	Rectal
Yokota et al. [37]	2008	RC	63	20.7	75.7	M, F	V	Y	Y	GI Pill, Rectal

**Table 3 sensors-22-07639-t003:** Comparison of the overarching design principles. One study did not report the sensor location. Model numbers prior to the parentheses indicate the number of models reporting the RMSE and found in Table 4 only, while the model numbers in parentheses include models listed in Table 4 and Table 5 reporting the RMSE, MD, and SEE. Values in italics indicate the outlier Tsadok et al. DW [47] was excluded from that data set.

Algorithm Design Principle	Number of Models	Total Number of Subjects (Reporting RMSE Only)	RMSE Mean ± SD
Heart Rate Only	4 (5)	139	0.32 ± 0.15
Heart Rate Plus Others	9 (11)	472	0.25 ± 0.13
No Heart Rate	3 (7)	39	0.29 ± 0.15
Chest Sensor Only	8 (9)	322	0.29 ± 0.14
Chest Sensor Plus Others	2 (4)	184	0.33 ± 0.14
No Chest Sensor	5 (9)	81	0.26 ± 0.15
Single Wearable Device	11 (15)	233	0.29 ± 0.15
Multiple Wearable Devices	4 (6)	354	0.29 ± 0.12

**Table 4 sensors-22-07639-t004:** Data on the model design and outcomes for models that reported the root mean square estimate (RMSE). NR: Not Reported and V = Variable.

Model	Year	Model Development Method	Number of Devices Worn	Device Locations	Sensor Types Used in Model	Number of Validation Subjects, *n*	RMSE	RMSE SE
Moyen et al. [31]	2021	Extended Kalman Filter	1	Upper Arm	HR, Skin Temp, Air Temp, Skin Humidity, Air Humidity, Accelerometry, Age, Height, Mass, Biological Sex	27	0.3	NR
Nazarian et al. [46]	2021	Kalman Filter	1	Wrist	HR, Skin Temp, Air Temp	15	0.13	NR
Buller et al. [58]	2020	Extended Kalman Filter with Sigmoid Curve	1	Chest	HR	21	0.43	0.20
Tsadok et al. CF [47]	2020	Cross Validation	1	Forehead	Heat Flux	13	0.23	0.1
Tsadok et al. CW [47]	2020	Cross Validation	1	Wrist	Heat Flux	13	0.31	0.14
Tsadok et al. DF [47]	2020	Drager Tcore	1	Forehead	Heat Flux	13	0.34	0.21
Tsadok et al. DW [47]	2020	Drager Tcore	1	Wrist	Heat Flux	13	1.97	1.26
Hunt et al. [59]	2019	Extended Kalman Filter	1	Chest	HR	8	0.32	NR
Laxminarayan et al. KF [56]	2018	Kalman Filter	2–11	Chest, Wrist, Variable	HR, Skin Temp, Activity Level	166	0.33	0.18
Laxminarayan et al. MM [56]	2018	Mathematical Model	2–11	Chest, Wrist, Variable	HR, Activity Level	166	0.33	0.18
Welles et al. [29]	2018	Kalman Filter	2	Chest	HR, Skin Temp, Heat Flux	4	0.18	0.04
Seng et al. [62]	2016	Extended Kalman Filter	1	Chest	HR, Skin Temp	5	0.29	NR
Buller et al. [64]	2015	Extended Kalman Filter	1	Chest	HR	27	0.21	0.11
Niedermann et al. [32]	2014	PCA with Multiple Linear Regression	6	Upper Arm, Lower Arm, Thigh, Chest, Back	HR, Skin Temp, Heat Flux	18	0.32	0.09
Buller et al. [34]	2013	Extended Kalman Filter	1	Chest	HR	83	0.3	0.13
Buller et al. [66]	2011	Kalman Filter with Dynamic Bayesian Network	1	Chest	HR, Heat Flux, Accelerometry	8	0.28	0.16
Yokota et al. [37]	2008	Compartmentalization	1	NR	HR, Air Temp, Air Humidity, Air Pressure, Wind Speed, Body Surface Area, Height, Mass	63	0.11	NR

**Table 5 sensors-22-07639-t005:** Data on the model design and outcomes for models that reported the mean difference or standard error of the estimate.

Model	Year	Model Development Method	Number of Devices Worn	Device Locations	Sensor Types Used in Model	*(n)*	MD	MD SE	SEE
Verdel et al. [57]	2021	Machine Learning (Proprietary)	1	Chest	HR, Skin Temp, Heat Flux	25	0.16	0.27	.
Eggenberger et al. MAX [60]	2018	PCA with Multiple Linear Regression	18	Chest, scapula, forearm, wrist, forearm, thigh, calf, hand, arm, sternum, rib, forehead	HR, Skin Temp (Insulated), Skin Temp (uninsulated)	6			0.278
Eggenberger et al. MIN [60]	2018	PCA with Multiple Linear Regression	2	Chest, scapula	HR, Skin Temp	6			0.29
Mazgaoker et al. [61]	2017	Drager Double Sensor	1	Forehead	Heat Flux	17	0.21	0.07	
Seo et al. [63]	2016	Kalman Filter	1	Chest	HR	27	0.26	0.4	
Richmond et al. [65]	2013	Bootstrap	1	Back of Neck	Skin Temp, Air Temp	32			0.2
Teunissen et al. [67]	2011	None	1	Aural Canal	Aural Temp	7	1.2	0.45	
Gunga et al. [33]	2008	Drager Double Sensor	1	Vertex of head	Heat Flux	20	0.2	0.75	

## Data Availability

The authors confirm that the data supporting the findings of this study are available within the article.

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
