# Peer review of "Wearable Sensor Technology to Predict Core Body Temperature: A Systematic Review"

_sensors, 2022, doi:10.3390/s22197639_

Round 1

Reviewer 1 Report

This is an interesting article comparing different measurement technologies for predicting core body temperature. I think this will be of interest to readers, especially in this day and age when environmental temperatures are getting higher every year. However, there are still some changes to be made before publication.

INTRODUCTION

- The first paragraph 46-57 is not quite appropriate for a scientific publication.

- The introduction should focus more on the technology and less on the HRI, since the goal of the paper is the technology and not the HRI. The technology is not mentioned until the fourth page.

RESULTS

- In the paper by Verdel et al. has variable exercisse conditions (10°C difference between Study 1 and Study 2). Please correct.

- I think that Figure 4, Table 4 and 5 are not part of the Demographic results.

- Table 6 should be at the beginning of the Results section (after Table 3).

- Is it possible to graphically present the results from Table 5 as well?

Author Response

·        Removed first paragraph of introduction.

·        Reorganization and abbreviation of introduction to focus more on the technology rather than HRI. Removed extraneous information about HRI and moved up information both about the importance of and measurement of CBT. Removed Table 1 from introduction as it focused exclusively on the symptoms, pathophysiology, and treatment of HRI whereas this paper is focused on detection and prevention, and it was leading to the technology being mentioned too late in the intro.

·        Corrected results for Verdel et al. by changing variable exercise conditions from N to Y.

·        Changed section headings of Results to better reflect the different results being presented. New headings read: Article Search Results, Aggregate Results, Results from Studies Reporting RMSE, Results from Studies Reporting MD or SEE.

·        Tables and figures have been appropriately reordered to align with new results headings, most notably Table 6 moved to become table 4. This includes placing formerly figure 4 and tables 4 and 5 within the new headings according to the results they displayed.

·        We chose not to graphically present the results from the table presenting MD and SEE results because there are only 4 and 3 algorithms for each, respectively, compared to 18 algorithms reporting RMSE and represented in Figure 4. The majority of the review focuses on RMSE results due to the ability to directly compare these results, as opposed to MD and SEE which have much fewer results to compare.

Reviewer 2 Report

The paper summarizes the current sensing technologies to predict body temperature, which is meaningful and it is a systematic investigation. The paper could be accepted with minor revision.

The format should be revised.  For instance, Method is 2. Method?

For the conclusion, table 7 is not a good format to summarize a paper.

Author Response

·        Corrected formatting at methods heading and throughout article.

·        Converted table 7 to figure 6, which more clearly presents the findings of the review and places them in the context of the HRI background and the suggested future directions for the technology. This conclusion is much more visual and easier to locate the desired information.

Reviewer 3 Report

The authors present an interesting review on a relevant and timely topic such as non-invasive wearable technologies able to estimate and predict core body temperature. The review is properly designed and conducted. The findings are well presented and message is clear. Overall, this review is of good quality and of interest for the readers. If I can suggest an additional point of discussion, it could be interesting to discuss how such technology which has been primarily designed and validated for sport and military contexts can be now translated to ore clinical setting as during the hospitalization. Indeed, despite the use of non-invasive/invasive core body temperature monitoring solutions are widely spread in ICU, in other wards temperature is often measured only at certain time points and with other estimates of body temperatures (e.g. tympanic temperature) and future studies should validate the technologies presented in this review in such clinical settings (e.g., Ajčević et al., Sensors 2022).

Author Response

  • The comment about adding a point on transition to clinic is a reasonable point but we did not feel it relevant enough to the discussion about accuracy and reliability in the field to add. As referenced in the introduction and table 2, there are a large number of CBT monitors validated for the clinical setting.

Round 2

Reviewer 1 Report

Authors stated: 

Corrected results for Verdel et al. by changing variable exercise conditions from N to Y.

However, in Table 2 it is still written N.

Author Response

Thank you for your review and attention to detail. We have made the following corrections upon final review of the data:

Verdel: Corrected variable exercise conditions and exercises to Y.

Laxminarayan, MIN and MAX models: changed clothing level to variable, corrected sensor types used in model and device locations. Renamed MAX and MIN models to KF and MM, respectively.

Eggenberger - Corrected variable exercise conditions to N

Updated aggregate counts in text in response to changes.
